# Mechanical Behaviour Evaluation of Full Iron Tailings Concrete Columns under Large Eccentric Short-Term Loading

**DOI:** 10.3390/ma16062466

**Published:** 2023-03-20

**Authors:** Xinxin Ma, Jianheng Sun, Fengshuang Zhang, Jing Yuan, Mingjing Yang, Zhiliang Meng, Yongbing Bai, Yunpeng Liu

**Affiliations:** 1Civil Engineering Department, Hebei Agricultural University, Baoding 071001, China; maxinxin1987@126.com (X.M.);; 2School of Materials Science and Engineering, University of Science and Technology Beijing, Beijing 100083, China

**Keywords:** iron tailings, RC column, large eccentric, short-term load, bearing capacity, ductility

## Abstract

In this study, full iron tailings concrete (FITC) was created using iron tailings from a tailings pond in Qian’an, China. Iron tailings account for 86.8% of the total mass of solid raw materials in the FITC. To enable large-scale use of FITC, a comprehensive investigation of the structural behaviour of full-iron tailing-reinforced concrete (FITRC) specimens is warranted. Therefore, eight rectangular reinforced concrete (RC) columns with conventional reinforced concrete (CRC) as a control were tested to investigate the effects of section dimensions, initial eccentricities, and concrete strengths, on the structural behaviour of FITRC columns under large eccentric short-term loading. The experimental and analytical results indicated that the sectional strain of the FITRC columns satisfied the plane-section assumption under short-term loading, and the lateral deflection curve agreed well with the half-sinusoidal curve. In addition, the FITRC columns exhibited a slightly lower cracking load and lower ultimate load capacity than the CRC columns, and the crack widths were larger than those of the CRC columns. The reduction in the load capacity observed in the FITRC was within the permissible range stated in the design code, thereby satisfying the code requirements. The deformation coefficients of the FITRC and CRC columns were identical, and the cracking and ultimate loads calculated according to the current code and theories were in good agreement with the measured results.

## 1. Introduction

Tailings are a major source of solid waste generated by the separation of valuable fractions from ore during mining operations [1,2]. Improper disposal of tailings occupies land resources, pollutes the environment, and releases hazardous substances [3]. In China, tailings are widely distributed [4] with a low utilisation rate [5]. Figure 1 shows photographs of piles of iron tailings. In 2018, 475 million tonnes of iron tailings were produced in China, accounting for 39.22% of the total tailings [6] (Figure 2). In 2019, 536 million tonnes of tailings were produced in China, of which 116 million tonnes were comprehensively utilised [7]. This represents a utilisation rate of 21.6%, which is significantly lower than the average utilisation rate in developed countries [8]. In addition, tailings dams pose a debris-flow hazard. Sudden failures of tailings dams occurred in Xiangfen County, Shanxi Province, China, in 2008 [9], and Jiaokou County, Shanxi Province, China, in 2022 [10]. These disasters injured hundreds of people and caused significant losses. Because of these problems, tailings have received considerable attention from the scientific community [11,12]. The optimal approach for dealing with the problems associated with iron tailings is to develop technologies for their large-scale use [13].

Concrete is a leading construction material, with an annual global consumption of approximately 25 billion tonnes [14]. The construction sector is responsible for over 30% of global greenhouse gas emissions [15]. In China, cement production is expected to reach approximately 2.38 billion tonnes in 2021 [16]. Several studies have recommended the use of tailings in the cement and concrete industry [17,18,19,20,21,22,23,24] to significantly reduce carbon dioxide emissions and consumption of natural resources [25].

Iron tailings are inert at room temperature [26,27]. Previous studies have used iron tailings as coarse [28,29] and fine aggregates [30,31,32,33,34,35], as well as silica materials for the production of aerated concrete at high temperatures [36,37,38]. Some studies have also used iron tailings as an admixture [39,40,41,42,43,44] and examined the activation of iron tailings powder [45,46]. The use of iron tailings in concrete has the potential to reduce the storage of iron tailings, reduce land use and reduce environmental pollution. At the same time, the replacement of fly ash with iron tailings powder as an admixture could reduce the cost of concrete and promote the sustainable development of concrete. In addition, iron tailings can improve the carbonation resistance, frost resistance, and sulphate erosion resistance of concrete [47,48].

To enable the extensive application of iron tailings in reinforced concrete, it is imperative to validate the structural behaviour of reinforced concrete samples containing iron tailings compared with plain concrete. A study was conducted to compare the flexural behaviour of iron tailings sand concrete beams and conventional concrete (CC) beams [49]. The results showed that the two types of concrete beams exhibited similar flexural behaviours and that the calculation of the crack width of iron tailings sand concrete beams should be corrected when the replacement rate of iron tailings sand is greater than 40%. Another study [50] investigated the axial compressive behaviour of short iron tailings sand concrete columns. The results showed that the iron tailings sand concrete short columns were comparable to CC short columns in terms of axial compressive strength, deformation, and ductility. The seismic behaviour of iron tailings sand concrete columns was investigated [51]. The results showed that the failure patterns of iron tailings sand concrete columns and CC columns were almost identical, while the flexural capacity of the iron tailings sand concrete columns was slightly different from that of the CC columns, and iron tailings sand could completely replace conventional sand. Compared with CC columns, the axial compressive properties of full iron tailing concrete columns satisfied the requirements of the current code [52]. Further, the calculation model that considered the hoop restraint effect could more accurately predict the axial compressive bearing capacity of full iron tailings concrete columns.

These studies show that the mechanical behaviour of reinforced concrete (RC) specimens made from iron tailings sand replacing fine aggregates is marginally different from that of conventional reinforced concrete (CRC) specimens. However, very few studies have been conducted on the mechanical behaviour of full iron tailings reinforced concrete (FITRC) specimens. The novelty of this study is the use of full iron tailings concrete (FITC), which was produced using iron tailings powder as an admixture, iron tailings gravel as a coarse aggregate, and iron tailings sand as a fine aggregate. To the best of our knowledge, this is the first investigation of the structural behaviour of FITRC columns under large eccentric loads. We aimed to comprehensively evaluate the characteristics and limitations of FITRC columns to provide a theoretical basis for the large-scale use of iron tailings. We believe that this study will be of practical significance to structural engineers in China. Previous dam failures in tailings ponds have caused significant casualties and property losses and have adversely affected the environment. Therefore, large-scale use of iron tailings in RC can effectively dispose of iron tailings, reduce the consumption of resources needed for cement production and sand mining, and ensure sustainable development.

## 2. Materials and Experimental Design

### 2.1. Materials

Table 1 shows the chemical compositions of the iron tailings, fly ash, river sand, and conventional gravel. Iron tailings are siliceous, with a silica content exceeding 60%.

All the samples were in powder form (<50 μm). The mineral composition of the raw materials was determined using X-ray diffraction (XRD). Quartz is the primary mineral in iron tailings, with trace amounts of anorthite, haematite, and microcline (Figure 3), all of which are inert materials that do not undergo hydration. This indicates that iron tailings have low cementitious reactivity in their original state. The particle sizes of the iron tailings were reduced, the original microstructure of the iron tailings was altered, and the reactivity of the iron tailings was improved by mechanical grinding. Cement consists of primary active cementitious materials, such as dicalcium silicate (C2S), tricalcium silicate (C3S), tricalcium aluminate (C3A), and tetracalcium aluminoferrite (C4AF). The main mineral constituents of fly ash are mullite, sillimanite, and quartz. The primary mineral constituents of river sand are cordierite, dolomite, cossyrite, and quartz. The primary mineral constituents of the conventional gravel are calcite, dolomite, and quartz.

Figure 4 shows the microscopic morphologies of the cementitious materials. The cement (Figure 4a) and iron tailings powder (Figure 4b) had varying particle sizes and many angularities. The fly ash particles were mostly spherical in shape (Figure 4c) with heterogeneous surfaces and showed some micro-porosities. Compared to iron tailings, fly ash can morphologically reduce the fluidity of concrete mixtures.

Ordinary Portland cement (P.O 42.5) was selected, and its primary indices are listed in Table 2. Iron tailings powder was sourced from Qian’an, Hebei, China, and mechanically activated by grinding in a single-shaft horizontal ball mill. Grade II fly ash was used. Table 3 lists the physical indices of the two mineral admixtures, and Figure 5 shows the particle size distributions of the two admixtures.

Table 4 summarises the primary indices of the iron tailings gravel and conventional gravel, showing that iron tailings gravel is superior to conventional gravel in terms of compressive strength and rock crushing index; however, the rate of expansion caused by the alkali–aggregate reaction of iron tailings gravel is inferior to that of conventional gravel. It is also superior to conventional gravel in terms of soundness. Iron tailings sand has a fineness modulus of 2.8, apparent density of 2780 kg·m^−3^ and bulk density of 1666 kg·m^−3^. River sand has a fineness modulus of 2.6, apparent density of 2650 kg·m^−3^ and bulk density of 1623 kg·m^−3^.

### 2.2. Experiment Design

The concrete mix proportions were designed according to the JGJ55-2011 specification in China [53]. In total, four concrete mix proportions and two strength grades were tested (grades C35 and C45). The specific proportions are listed in Table 5. Considering that fly ash can reduce the fluidity of concrete mix, FITC is superior to CC in terms of workability at the same water/binder ratio and water-reducing agent dosage. The mass of the iron tailings was 87.0% of the total mass of the FITC.

The mechanical behaviour of the concrete, including cubic and prismatic compressive strength, splitting tensile strength, modulus of elasticity, and Poisson’s ratio, was tested according to GB/T 50081-2019 [54]. The test results are presented in Table 6. The cubic, prismatic compressive, and tensile strengths of FITC were slightly lower than those of CC, and the modulus of elasticity of FITC decreased by 19.1% and 18.4%, respectively, compared with CC.

An HRB400 grade rebar was selected, consisting of 14 mm and 16 mm diameter longitudinal bars and 8 mm diameter stirrups. Figure 6 shows the stress–strain curves of the rebars, and Table 7 lists the primary engineering indices of the rebars. Symmetrical and asymmetrical reinforcements were used for the RC columns. Figure 7a shows the details of the reinforcement. Two axial linear variable differential transformers (LVDTs) and five lateral LVDTs were placed on the RC columns, as shown in Figure 7.

To mitigate the effect of an additional bending moment on an eccentrically pressurised column specimen, the span-depth ratio (*L*/*h*) should not exceed 5. The 1200 mm high specimen had 150 mm × 250 mm sectional dimensions and symmetrical reinforcement of two 14 mm rebars in tension and two 14 mm rebars in compression made of C45 strength concrete. The 1500 mm-high specimen had a 200 mm × 300 mm section and asymmetric reinforcement with three 16 mm rebars in tension and two 16 mm rebars in compression made from C35 concrete. To ensure that the specimens could withstand a large eccentric load, cow legs were placed at the top and bottom of the specimen. The concrete was poured into three FITRC columns and one CRC column.

Before the start of the experiment, the RC columns were placed at the location specified by the initial eccentricity *e*_0_ and preloaded. The preload did not exceed 20% of the ultimate load, and the column stability was evaluated. The columns were formally loaded in several stages of 5%–15% of the calculated ultimate bearing capacity *N_u_*. Once the specified load was reached after each loading stage, the load was maintained for 3 min to fully release the strain in the RC, and the strain and displacement were measured at the corresponding time points.

## 3. Experimental Results and Discussion

### 3.1. Failure Modes and Crack Propagation

The failure modes of the FITRC and CRC columns under large eccentric loads were similar. The failure of all specimens was manifested by longitudinal bar yielding in the tension zone, followed by concrete crushing in the compression zone (Figure 8). The eccentricity of the FITRC45 and CRC45 columns was 0.60 and that of the FITRC35 and CRC35 columns was 0.70. In a previous study [55], the eccentric compressive behaviour of iron tailings sand RC columns was investigated, and similar results were obtained.

Cracks occurred in the tension zone at the midspan point along the lateral depth of the FITRC45 and CRC45 columns when the load reached approximately 0.2 *N_u_* and propagated with increasing load, resulting in new microcracks. When the load reached approximately 0.9 *N_u_*, vertical cracks were observed in the lateral compression zone of the column specimens and began to propagate. When the ultimate load *N_u_* was reached, the concrete cracks in the compression zone elongated and were accompanied by the appearance of new microcracks. Some of the concrete was crushed as the test specimen continued to be loaded, and the ultimate compressive strain of the concrete in the compression zone was reached; thus, the axial load on the column specimens decreased. Similar failure processes were observed for FITRC35 and CRC35 columns.

Regardless of whether the eccentricity was 0.60 or 0.70, the final failure of all the RC columns was demonstrated by the crushed concrete in the outer layer of the stirrups and the intact concrete in the inner layer of the stirrups. This can be attributed to the fact that the stirrups significantly restrained the concrete in the inner layer and inhibited its deformation. In addition, Figure 8 shows that the area of crushed concrete was related to the eccentricity of the specimen section; that is, a smaller eccentricity indicated a larger area of crushed concrete.

Figure 9 shows the load–maximum crack width curves of the RC columns. The crack width propagation trends of FITRC and CRC columns remained similar for eccentricities of 0.60 and 0.70, and the curves of the four RC columns overlapped. Considering that the splitting tensile strength of FITC was lower than that of CC, the crack width of FITRC columns was larger than that of CRC columns at the same load level.

### 3.2. Load–Deflection Relationships

Figure 10 shows the axial load–deflection relationship (axial and lateral displacements) of RC columns under large eccentric loads. Axial and lateral displacements were measured using a vertical displacement gauge and a displacement gauge at the midspan point along the column depth, respectively. The negative and positive values in the figure indicate the axial and lateral displacements, respectively, at the midspan point along the column depth. Prior to longitudinal bar yielding, there was an approximately linear relationship between the loads and displacements of all the RC columns. After tensile yielding of the longitudinal bars, the stiffness of the RC columns decreased and the load-deflection curve increased nonlinearly until the peak load was reached when the concrete cover was crushed and spalled. After the peak load, the curve shows a decreasing trend; the load of the RC columns showed a rapid reduction of 5–10% followed by a more gradual reduction, and ductile failures occurred in the specimens. The vertical and lateral displacements of the CRC columns were clearly smaller than those of the FITRC columns. In particular, there were smaller lateral displacements compared to vertical displacements, considering that the modulus of elasticity of CC was approximately 19% greater than that of FITC (Table 6). Similarly, the peak loads of the CRC columns were slightly higher than those of the FITRC columns (Table 8).

Figure 11 shows the lateral displacements of four typical RC columns distributed along the column depth (measured using five displacement gauges). The RC columns exhibited similar lateral displacements at the different loading stages. Furthermore, the columns showed slow and rapid increases in lateral displacement in the early loading stages and as the peak load approached, respectively. The lateral deformations of the RC columns were caused by first- and second-order moments. In accordance with the literature [56], the lateral deformations and sinusoidal waveforms of columns hinged at both ends remained similar. Therefore, the lateral deformation of the RC columns can be expressed as follows:(1)$$D_{L}=\Delta_{p}\sin\left(\frac{\pi \cdot l}{L}\right)$$
where Δ*_p_* is the maximum displacement at the midspan point along the column depth, *l* is the location along the column depth, *L* is the column depth, and *D_L_* is the lateral displacement of the column at l. Figure 11 compares the lateral deformations of the RC columns and the sinusoidal model, which agree well for eccentricities of both 0.60 and 0.70, thereby indicating that the sinusoidal model can effectively predict the lateral deformations of FITRC columns at different loading stages.

### 3.3. Deformation and Ductility

Table 8 lists the primary test results of the RC columns, including the peak load *N_u_*, displacement at the corresponding peak load Δ*_p_*, and yield displacement Δ*_y_*. The behaviour parameters of the RC columns were quantified to determine the deformation capacity of the RC columns. In seismic design, the inelastic deformation capacity of specimens is generally quantified using the displacement ductility factor [57] and deformation coefficient [58]. The displacement ductility factor indicates the ductility behaviour of the specimens, which can be calculated as follows.
(2)$$\mu =\frac{\Delta_{0.85}}{\Delta_{y}}$$

The deformation coefficient indicates the deformation capacity of the specimens after reaching the peak load and can be calculated as follows.
(3)$$\lambda =\frac{\Delta_{0.85}}{\Delta_{p}}$$

Here, Δ_0.85_ is the corresponding displacement at *N_u_* of 0.85 in the load decreasing stage [57], Δ*_p_* is the peak lateral deflection at the peak load, and Δ*_y_* is the corresponding displacement at the yield load when the limit is reached in the elastic stage. The yield displacement Δ*_y_* can be obtained using a graphical method [57] (Figure 12).

Table 8 shows that the mean ductility factors of the FITRC45 and FITRC35 columns were 1.78 and 2.82, respectively, and the ductility factors relative to the CRC45 and CRC35 columns were reduced by 36.6% and 19.5%, respectively. Therefore, an increase in section size and eccentricity reduced the difference in ductility between FITRC35 and CRC35 columns. A previous study [52] examined the axial compressive properties of full iron tailings RC columns and also found that the ductility coefficients of FITRC columns were lower than those of CRC columns. The mean deformation coefficients of the FITRC45 and FITRC35 columns were 1.33 and 1.35, respectively, and the deformation coefficients relative to CRC45 and CRC35 columns were reduced and increased by 5.27% and 2.80%, respectively. In the load-decreasing phase, the deformation capacities of FITRC and CRC columns were identical.

As can be seen from Table 8, the lateral deflection Δ*_p_* corresponding to the peak load of the FITRC columns was greater than that corresponding to the peak load of the CRC columns, and the mean values of Δ*_p_* of the FITRC45 and FITRC35 columns increased by 25% and 21%, respectively, compared to the CRC45 and CRC35 columns.

### 3.4. Load–Strain Relationships

Figure 13 shows the load–rebar strain curves of typical RC columns, where *c* is *A_sc_* of the compressive rebars, which were close to the axial compressive force with a negative strain value, and *t* is *A_st_* of the tensile rebars, which were away from the axial compressive force (Figure 7) with a positive strain value. ε*_y_*_14_ and *ε_y_*_16_ are the tensile yield strains of rebars with diameters of 14 and 16 mm, respectively. Figure 13 shows that the strains of *A_st_* in the RC columns were greater than ε*_y_*_14_ and *ε_y_*_16_, and the yield strain of rebars in the compression zone was reached before the peak load was reached. The stress state is a typical feature of compression failure at large eccentricities. In addition, the strain of rebars in the FITRC columns was significantly higher than that in the CRC columns for both *A_sc_* and *A_st_*, because FITC had a lower modulus of elasticity than CC. In addition, the prismatic compressive and cracking strengths of FITC were slightly lower than those of CC. As a result, the ultimate load of the FITC columns was lower than that of the CRC columns, while the lateral deflection and axial displacement of the FITC columns are greater than those of the CRC columns.

Concrete strain gauges were placed along the section depth of the RC column, as shown in Figure 7a, to study the strain distribution of the concrete during loading. Figure 14 shows the strain distribution of typical concrete. Under large eccentric loads, the concrete in the tension zone of the RC columns cracked, resulting in failure of the concrete strain gauges near the tension zone. In particular, the strain distributions in the concrete before and after cracking were recorded. Figure 13 shows that the concrete strain was linearly distributed along the depth of the RC column section. Therefore, FITRC columns satisfied the planar section assumption, and the flexural capacity of FITRC columns can be theoretically calculated according to the planar section assumption.

## 4. Analysis of Sectional Capacities

### 4.1. Moment Magnification Factor

Considering the axial and flexural deformations that occurred in the RC columns under large eccentric loads during the experiment, the axial capacity, moment, and crack resistance should be calculated across the section at the mid-span of the column depth. Figure 15 shows a schematic of the second-order effects of the RC columns. Under the eccentric axial load N with initial eccentricity *e*_0_, the lateral displacement Δ*_p_* of the RC columns across the column depth at the midspan allowed the eccentricity of the axial load relative to the centre of mass of the section to reach Δ*_p_* + *e*_0_. Meanwhile, the moment of the RC columns increased from *M*_1_ = *Ne*_0_ to *M_max_* = *M*_1_ + *M*_2_ = *N* (*e*_0_ + Δ*_p_*), which is known as the second-order effect of eccentrically loaded columns [59,60]. During design, the second-order effects are accounted for using the moment augmentation factor [61].

According to [62], the relationship between the ultimate sectional curvature *φ_p_* and Δ*_p_* of RC columns can be expressed as:(4)$$\varphi_{p}=\Delta_{p}{\left(\frac{\pi }{L}\right)}^{2}$$

This equation is valid only if the first- and second-order deformations of a column can be expressed as sinusoidal shapes.

Therefore, the moment augmentation factor *η* can be expressed as:(5)$$\eta =\frac{e_{0}+\Delta_{p}}{e_{0}}=1+\frac{\varphi_{p}L^{2}}{e_{0}\pi^{2}}$$

In the design code, an additional eccentricity *e_a_* is introduced because of uncertain load locations, uneven concrete quality, and construction variations, and a value of 20 mm is considered with an eccentricity of *e_i_* = *e*_0_ + *e_a_*.

### 4.2. Bearing Capacity

In general, the bearing capacity under eccentric compression is calculated using the equivalent rectangular stress diagram [61], assuming that the FITRC columns satisfy the plane section assumption, and the theoretical calculation can be performed according to GB 50010-2010 [61]. Furthermore, to simplify the calculation process, the tensile strength of the concrete was ignored. Figure 16 shows a simplified diagram for calculating the section of an RC column subjected to compression failure under a large eccentric load.

The compressive stress curve of the concrete in the compression zone was replaced with an equivalent rectangular diagram. According to the equilibrium conditions of the forces, the following formula was obtained.
(6)$$N_{u}=\alpha_{1}f_{c}bx+f_{yc}A_{sc}-f_{yt}A_{st}$$

Here, *α*_1_ is the equivalent rectangular stress block coefficient of concrete in the compression zone, defined as 1.0; *b* and *h* are the sectional dimensions of the RC columns; *f_yt_* and *f_yc_* are the yield strengths of the tensile and compressive longitudinal bars, respectively; *A_st_* and *A_sc_* are the sectional areas of the tensile and compressive longitudinal bars, respectively. According to the resultant point of various forces on the tensile specimens in Figure 16b, the moment equilibrium conditions were determined and expressed as:(7)$$N_{u}e=\alpha_{1}f_{c}bx\left(h_{0}-0.5x\right)+f_{yc}A_{sc}\left(h_{0}-a_{sc}\right)$$
(8)$$e=\eta e_{i}+0.5h-a_{st}$$
where *e* is the distance from the point of axial force to the resultant point of *A_st_* of tensile longitudinal bars, *h*_0_ is the distance from the resultant point of *A_st_* of tensile longitudinal bars to the edge of the compressive concrete, *x* is the depth of concrete in the compression zone, *a_st_* is the distance from the resultant point of *A_st_* of tensile longitudinal bars to the edge of the tensile concrete, and *a_sc_* is the distance from the resultant point of *A_sc_* of compressive longitudinal bars to the edge of the compressive concrete.

### 4.3. Crack-Resistant Load

According to the SL191-2008 standard in China [63], the crack resistance of RC columns under eccentric compression should be calculated as follows:(9)$$N_{cr}\leq \frac{\gamma_{m}\alpha_{ct}f_{t}A_{0}W_{0}}{e_{i}A_{0}-W_{0}}$$
where *γ_m_* is the plastic section modulus, with the rectangular section set to 1.55; *α_ct_* is the tensile stress limit coefficient of concrete, offset to 0.85; *A*_0_ is the area of the transformed section; and *W*_0_ is the elastic section modulus of the tensile edge of the transformed section, calculated as:(10)$$A_{0}=bh+\alpha_{E}A_{st}+\alpha_{E}A_{sc}$$
(11)$$W_{0}=I_{0}/(h-y_{0})$$
(12)$$I_{0}=(0.0833+0.19\alpha_{E}\rho )bh^{3}$$
(13)$$y_{0}=(0.5+0.425\alpha_{E}\rho )h$$
where *I*_0_ is the moment of inertia of the column section, *y*_0_ is the distance from the axis of gravity of the transformed section to the compression edge, and *ρ* is the reinforcement ratio of the tensile longitudinal bars.

### 4.4. Experimental Verification of Theoretical Predictions

According to the aforementioned formulae, the moment augmentation factor and bearing capacity of the RC columns were calculated using the measured mechanical parameters of the rebar and concrete, respectively, and compared with the experimental results. As shown in Table 9, the theoretically calculated cracking load and ultimate bearing capacity were in good agreement with the measured results, with a maximum error of only 16%. In addition, there was a strength safety margin, which verifies that the calculation formulae are effective for FITRC columns under large eccentric loads.

Table 10 lists the moments of the RC columns and the calculated moment augmentation factor. According to the structural design code [61], the moment augmentation factor *η* should be 1.0 when *L*/*h* ratios do not exceed 5.0. Furthermore, the moment augmentation factor of the FITRC columns is greater than that of the CRC columns in the control group. The measured moment augmentation factors of the six FITRC columns are close to each other, with a maximum value of 1.079, whereas the *η* value of the FITRC columns increases by a maximum of 1.5% compared to the CRC columns. This is because the *L*/*h* ratios of the RC columns in this study do not exceed 5.0 and the impact of the second-order effects is marginal.

In addition, Table 10 shows that the second-order moments of the FITRC columns are larger than those of the CRC columns. Furthermore, the average ratio of the second-order moments of the three FITRC45 columns to the CRC45 columns is 1.14, and that of the three FITRC35 columns to the CRC35 columns is 1.15. Therefore, if the second-order effect is to be considered in the structural design, the moment augmentation factor *η* of the FITRC columns should be 1.15 for safety reasons.

## 5. Conclusions

This study investigated the structural behaviour of FITRC columns under large eccentric loading. Six FITRC columns and two CRC column specimens were examined to investigate the effect of different raw materials, section dimensions, and eccentricities, on the mechanical behaviour of RC columns under large eccentric compression. The following conclusions were drawn from this study:Under large eccentric short-term loads, the failure modes of the FITRC and CRC columns were found to be similar, and the failures were manifested by the yielding of the tensile and compressive rebars and concrete crushing in the compression zone. The long-term behaviour of FITRC (creep and shrinkage) requires further investigation.As the prismatic compressive and tensile strengths of the FITRC columns were slightly lower than those of the CRC columns, the ultimate load capacity of the FITRC columns was slightly lower than that of the CRC columns, and the crack widths of the FITRC columns were greater than those of the CRC columns.The sectional strain of the FITRC columns, which was similar to that of the CRC columns, satisfied the planar section assumption, and the lateral deflection curve agreed well with the half-sinusoidal assumption, which is consistent with the CRC column assumption in the current specifications.Compared with the CRC45 and CRC35 columns, the ductility factors of the FITRC45 and FITRC35 columns were 36.6% and 19.5% lower, respectively. The underlying cause of this phenomenon was the comparatively low modulus of elasticity of FITC, which resulted in a more pronounced lateral deformation of the FITRC columns when subjected to eccentric loading than the CRC columns under equivalent conditions.Based on the current Chinese standards, the theoretical calculations for the cracking load and ultimate load capacity of FITRC columns are relatively accurate. The calculation results indicate that FITRC columns have a certain safety reserve, and that FITC has the potential for practical application in the construction sector.Because the lateral deflection of the FITC columns was greater than that of the CRC columns, the second-order moments of the FITC columns were greater than those of the CRC columns. If the second-order effect is considered in the structural design, the moment augmentation factor of the FITRC columns should be 1.15 for safety reasons. Therefore, FITRC columns with high *L*/*h* ratios should be investigated further.

The ductility factor of FITRC columns is much lower than that of CRC columns, so the ductility of FITRC columns can theoretically be improved by stirrup reinforcement. However, this hypothesis requires further investigation.

## Figures and Tables

**Figure 1 materials-16-02466-f001:**
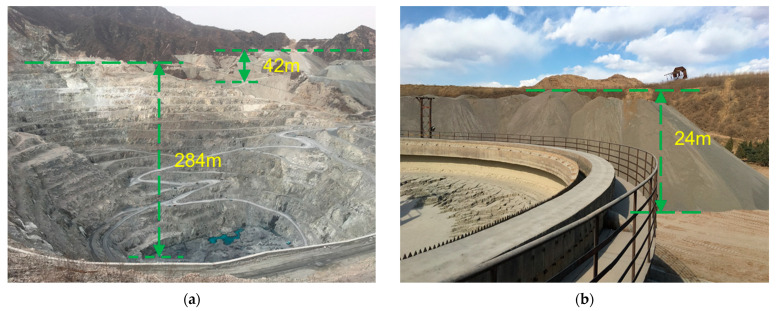
Photos of iron tailings stacking dam in Hebei, China: (**a**) mining and tailings site; (**b**) tailings site.

**Figure 2 materials-16-02466-f002:**
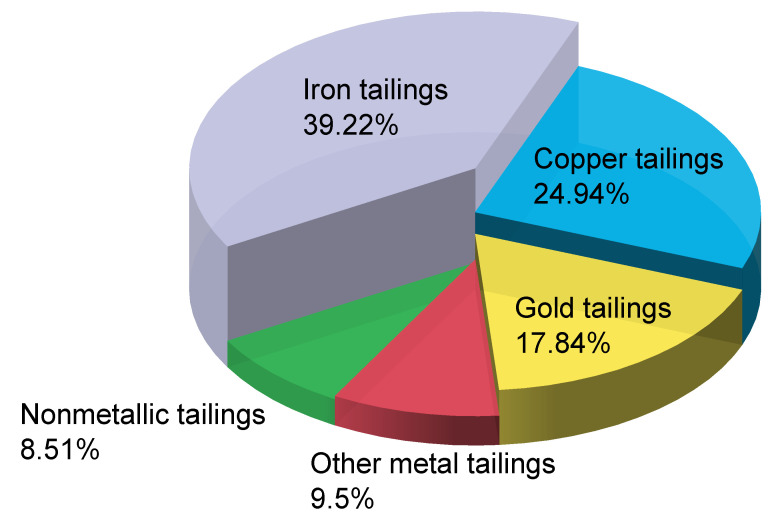
Breakdown of tailings production in 2018.

**Figure 3 materials-16-02466-f003:**
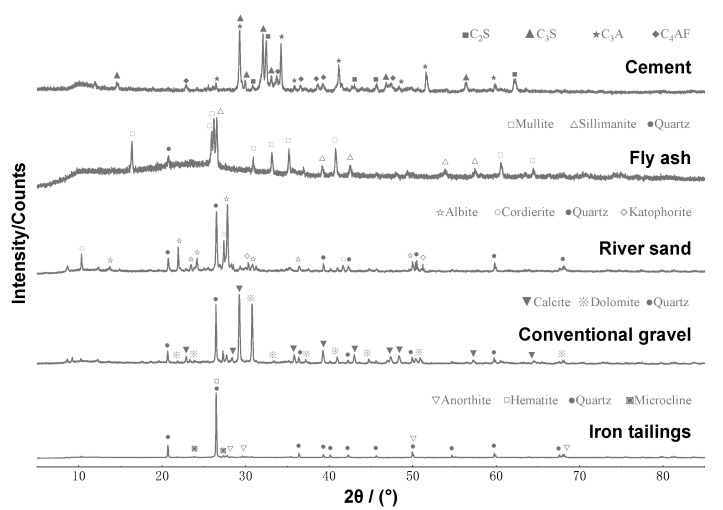
XRD patterns of concrete raw materials.

**Figure 4 materials-16-02466-f004:**
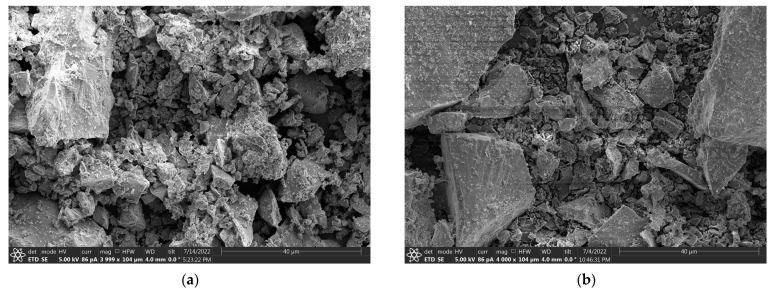
SEM micrograph at 4000× magnification of particles: (**a**) cement; (**b**) iron tailings; (**c**) fly ash.

**Figure 5 materials-16-02466-f005:**
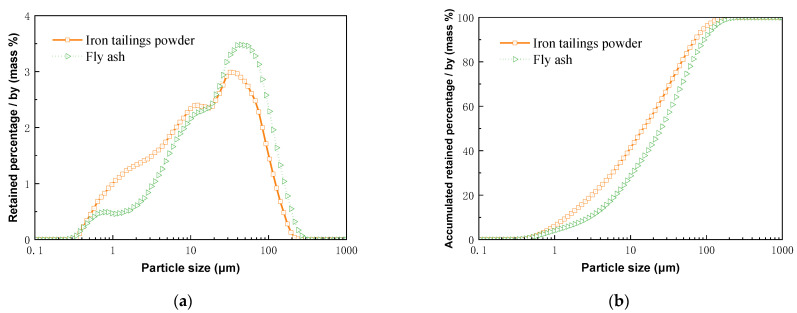
Particle size distribution curves of two mineral admixtures: (**a**) particle size distribution; (**b**) cumulative particle size distribution.

**Figure 6 materials-16-02466-f006:**
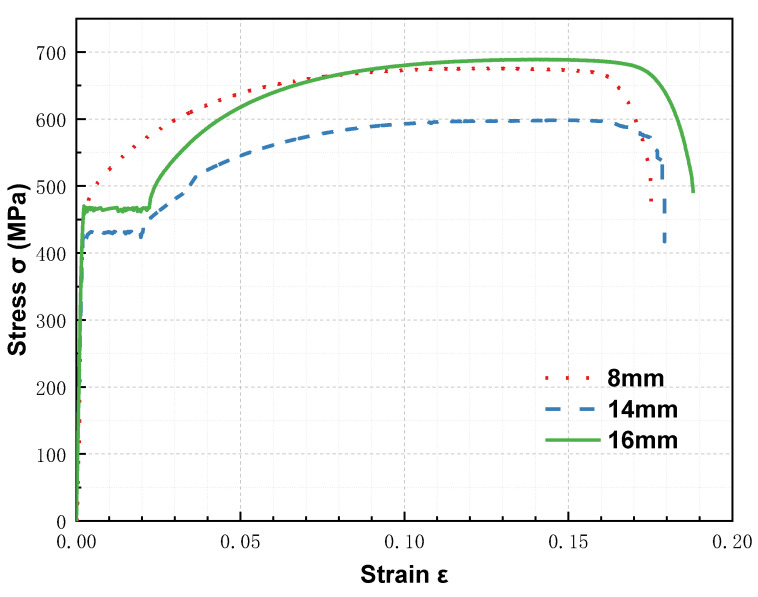
Stress–strain curve of rebars.

**Figure 7 materials-16-02466-f007:**
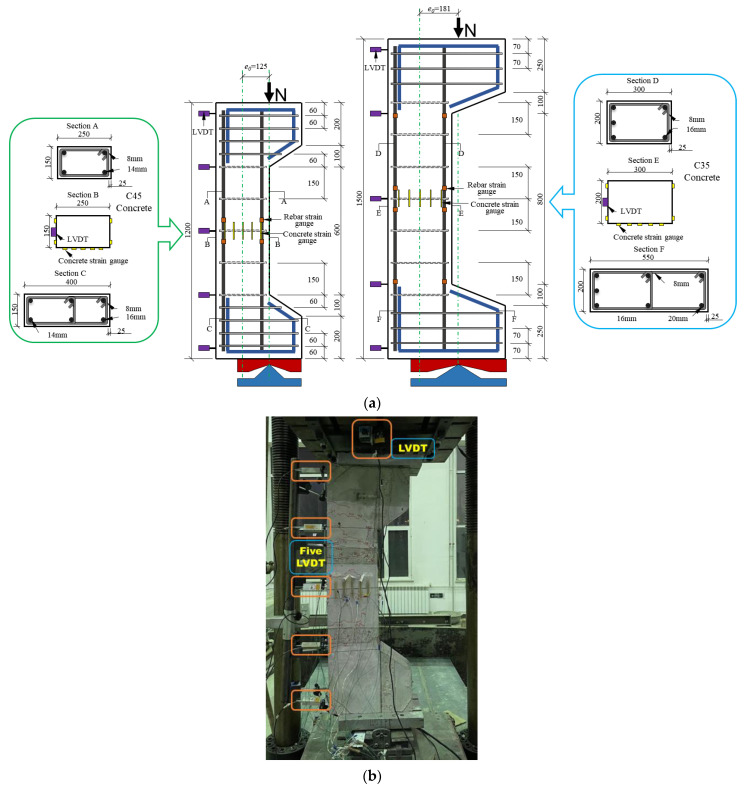
Compression test on RC columns under large eccentric loading: (**a**) geometry and reinforcement details of RC columns (unit: mm); (**b**) test setup.

**Figure 8 materials-16-02466-f008:**
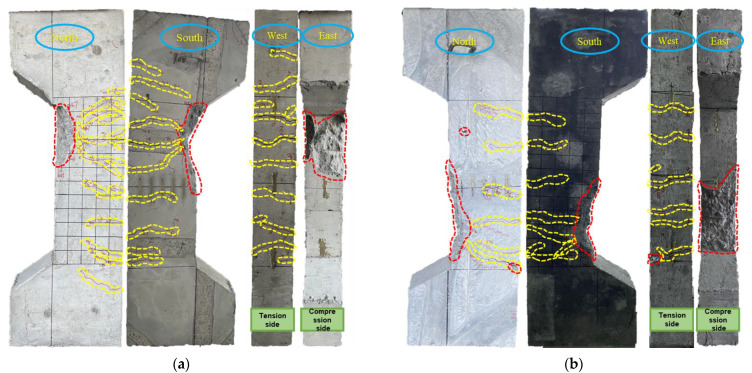
Typical failure modes of RC columns: (**a**) FITRC45-2 column specimen; (**b**) CRC45 column specimen; (**c**) FITRC35-2 column specimen; (**d**) CRC35 column specimen.

**Figure 9 materials-16-02466-f009:**
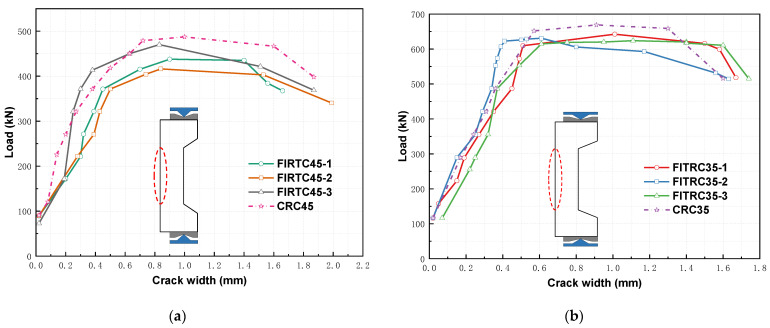
Load–maximum crack width curves of RC columns: (**a**) *e*_0_/*h*_0_ = 0.60; (**b**) *e*_0_/*h*_0_ = 0.70.

**Figure 10 materials-16-02466-f010:**
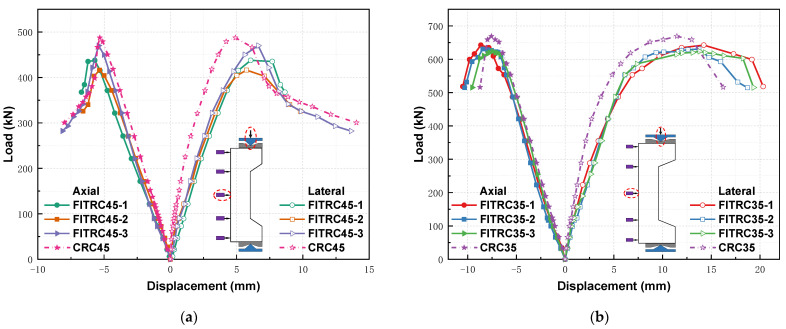
Load–deflection curves of RC columns: (**a**) *e*_0_/*h*_0_ = 0.60; (**b**) *e*_0_/*h*_0_ = 0.70.

**Figure 11 materials-16-02466-f011:**
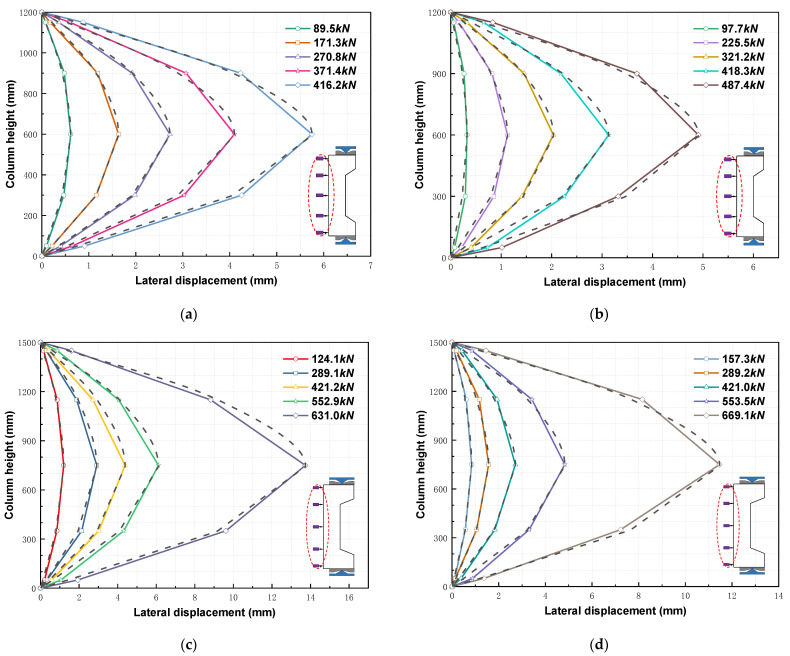
Lateral deformation along the section depth of typical RC columns: (**a**) FITRC45-2 column specimen; (**b**) CRC45 column specimen; (**c**) FITRC35-2 column specimen; (**d**) CRC35 column specimen.

**Figure 12 materials-16-02466-f012:**
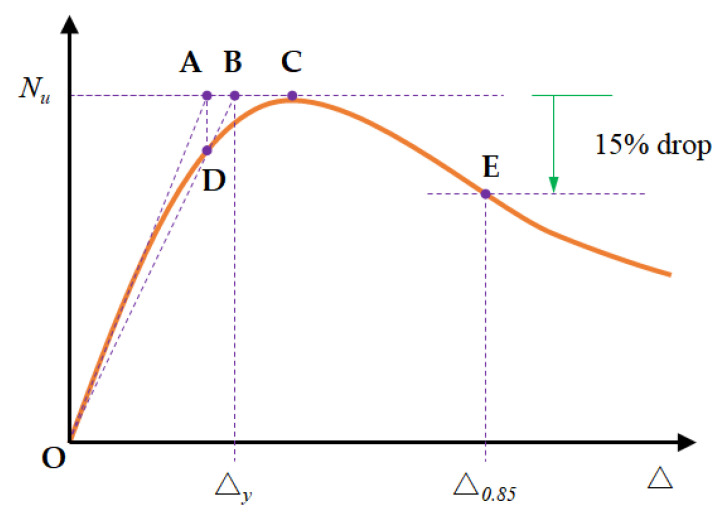
Definitions of yield displacement Δ*_y_* and ultimate displacement Δ_0.85_.

**Figure 13 materials-16-02466-f013:**
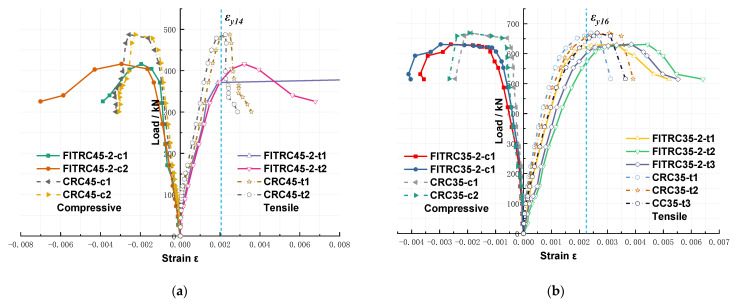
Load–rebar strain curves of typical RC columns: (**a**) *e*_0_/*h*_0_ = 0.60; (**b**) *e*_0_/*h*_0_ = 0.70.

**Figure 14 materials-16-02466-f014:**
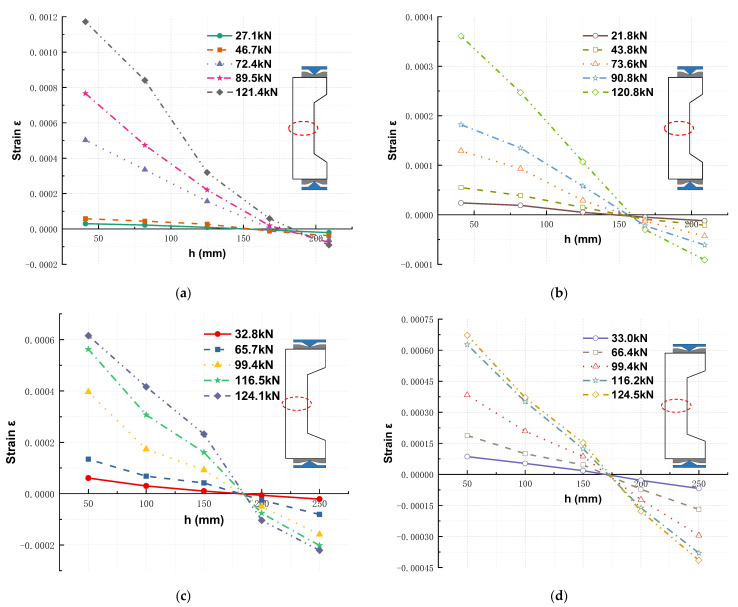
Typical concrete strain distribution along the section depth at the mid-span point: (**a**) FITRC45-2 column specimen; (**b**) CRC45 column specimen; (**c**) FITRC35-2 column specimen; (**d**) CRC35 column specimen.

**Figure 15 materials-16-02466-f015:**
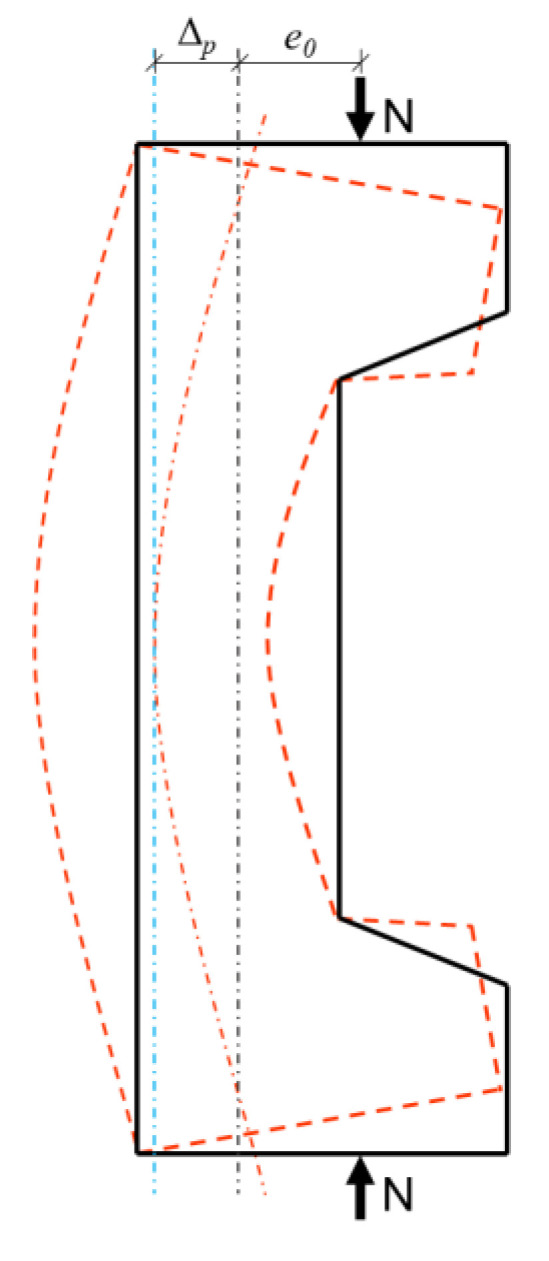
Schematic diagram of the second-order effect.

**Figure 16 materials-16-02466-f016:**
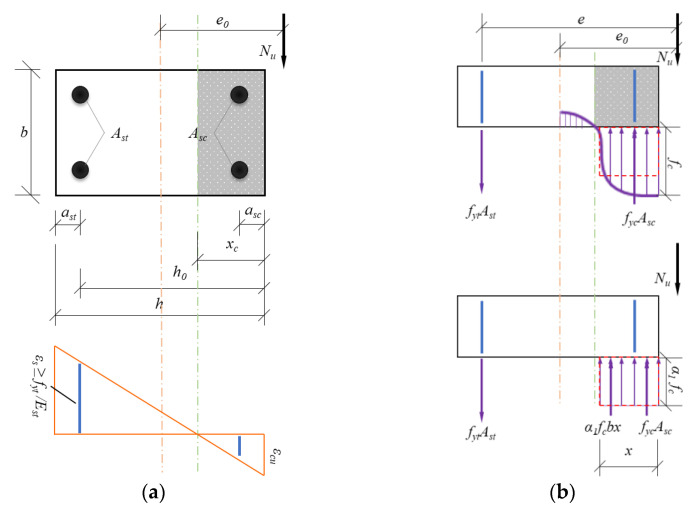
Simplified diagram for calculation of the section subjected to compression failure under large eccentric loading: (**a**) strain diagram of the specimens; (**b**) equivalent calculation diagram.

**Table 1 materials-16-02466-t001:** Main chemical composition of concrete raw materials.

Specimens	SiO_2_	Fe_2_O_3_	MgO	Al_2_O_3_	CaO	CaCO_3_
Iron tailings	68.2%	12.5%	6.8%	5.1%	4.8%	0.0%
Fly ash	52.2%	5.6%	1.3%	26.5%	4.2%	0.0%
River sand	44.0%	3.3%	16.0%	6.4%	27.5%	0.0%
Conventional gravel	15.0%	0.3%	16.2%	0.2%	0.0%	68.1%

**Table 2 materials-16-02466-t002:** Primary technical indices of Portland cement.

Apparent Density	Flexural Strength (MPa)		Compressive Strength (MPa)	
(kg·m^−3^)	3 d	28 d	3 d	28 d
3090	6.9	10.8	31.2	52.7

**Table 3 materials-16-02466-t003:** Primary technical indices of mineral admixtures.

Specimens	D_10_ (μm)	D_50_ (μm)	D_90_ (μm)	SSA (m^2^·kg^−1^)	AD (m^2^·kg^−1^)
Iron tailings powder	1.51	15.51	74.84	480.6	2770
Conventional gravel	3.02	26.51	98.18	415.1	2180

**Table 4 materials-16-02466-t004:** Main technical indices of coarse aggregate.

Specimens	Compressive Strength of Rock (MPa)	Crushing Index (%)	Soundness (%)	Alkali–Aggregate Reaction (%)	AD (m^2^·kg^−1^)	BD (kg·m^−3^)
Iron tailings gravel	62.2	6.05	3.5	0.051	2730	1570
Conventional gravel	60.0	6.56	3.7	0.042	2700	1550

**Table 5 materials-16-02466-t005:** Design of mix proportions.

Specimens	Water Binder Ratio	Water (kg)	Cement (kg)	Iron Tailings Powder (kg)	Iron Tailings Sand (kg)	Iron Tailings Gravel (kg)	Fly Ash (kg)	River Sand (kg)	Conventional Gravel (kg)	Water Reducing Agent (kg)	Slump (mm)	Dispersion (mm)
FITC45	0.32	170	369	158	739	1017	-	-	-	3.1	210	520 × 530
CC45	0.32	170	369	-	-	-	158	706	976	3.1	200	470 × 490
FITC35	0.40	170	301	129	775	1071	-	-	-	2.5	215	515 × 535
CC35	0.40	170	301	-	-	-	129	746	1031	2.5	205	475 × 490

**Table 6 materials-16-02466-t006:** Mechanical behaviour of concrete.

Specimens	Cubic Compressive Strength *f_cu_* (MPa)	Prismatic Compressive Strength *f_c_* (MPa)	Splitting Tensile Strength *f_t_* (MPa)	Static Modulus of Elasticity *E_c_* (MPa)	Poisson’s Ratio *ν*
FITC45	53.5	38.1	3.5	3.22 × 10^4^	0.27
CC45	55.9	41.4	3.7	3.98 × 10^4^	0.24
FITC35	44.4	34.8	3.2	3.05 × 10^4^	0.25
CC35	47.8	36.3	3.4	3.74 × 10^4^	0.23

**Table 7 materials-16-02466-t007:** Mechanical behaviour of rebars.

Reinforcement Diameter (mm)	Yield Strength *f_y_* (MPa)	Ultimate Strength *f_u_* (MPa)	Elastic Modulus *E_s_* (MPa)	Percentage Elongation after Fracture (%)
8	460	676	2.06 × 10^4^	17.2
14	430	598	2.09 × 10^4^	19.0
16	466	689	2.07 × 10^4^	20.3

**Table 8 materials-16-02466-t008:** Main performance indices of RC columns.

Specimens	*N_u_* (kN)	Δ*p* (mm)	Δ*y* (mm)	Δ_0.85_ (mm)	μ	λ
FITRC45-1	437.7	6.06	5.23	8.04	1.54	1.33
FITRC45-2	416.2	5.75	4.00	8.58	2.15	1.49
FITRC45-3	470.1	6.63	4.71	7.75	1.65	1.17
CRC45	487.4	4.91	2.46	6.89	2.80	1.40
FITRC35-1	642.5	14.23	6.62	19.80	2.99	1.39
FITRC35-2	631.0	13.73	6.72	17.32	2.58	1.26
FITRC35-3	624.0	13.80	6.60	19.17	2.90	1.39
CRC35	669.1	11.46	4.28	15.02	3.51	1.31

**Table 9 materials-16-02466-t009:** Bearing capacity of RC columns.

Specimens	*N_cr_* (kN)	*N_u_* (kN)	*N_cr-t_* (kN)	*N_u-t_* (kN)	$\frac{N_{cr}}{N_{cr-t}}$	$\frac{N_{u}}{N_{u-t}}$
FITRC45-1	89.5	437.7	85.3	405.2	1.05	1.08
FITRC45-2	89.5	416.2	85.3	405.2	1.05	1.03
FITRC45-3	90.3	470.1	85.3	405.2	1.06	1.16
CRC45	90.8	487.4	86.8	418.6	1.05	1.16
FITRC35-1	116.5	642.5	110.8	598.9	1.05	1.07
FITRC35-2	116.5	631.0	110.8	598.9	1.05	1.05
FITRC35-3	116.3	624.0	110.8	598.9	1.05	1.04
CRC35	116.2	669.1	111.9	608.4	1.04	1.10

**Table 10 materials-16-02466-t010:** Moments of RC columns.

Specimens	*N_u_* (kN)	Δ*_p_* (mm)	*η*	*M*_1_ (kN·m)	*M*_2_ (kN·m)	*M_max_* (kN·m)	$\frac{M_{2}}{M_{2}(CRC)}$
FITRC45-1	437.7	6.06	1.048	54.7	2.7	57.4	1.11
FITRC45-2	416.2	5.75	1.046	52.0	2.4	54.4	1.00
FITRC45-3	470.1	6.63	1.053	58.8	3.1	61.9	1.30
CRC45	487.4	4.91	1.039	60.9	2.4	63.3	1.00
FITRC35-1	642.5	14.23	1.079	116.3	9.1	125.4	1.19
FITRC35-2	631.0	13.73	1.076	114.2	8.7	122.9	1.13
FITRC35-3	624.0	13.80	1.076	112.9	8.6	121.6	1.12
CRC35	669.1	11.46	1.063	121.1	7.7	128.8	1.00

## Data Availability

Not applicable.

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
