# Peer review of "Mechanical Behaviour Evaluation of Full Iron Tailings Concrete Columns under Large Eccentric Short-Term Loading"

_materials, 2023, doi:10.3390/ma16062466_

Round 1

Reviewer 1 Report

The article is a collection of interesting results.

The authors deliver results in large quantities, but what is lacking is to highlight what these results bring back that is innovative compared to what exists in the bibliography. Authors are encouraged to compare their results with other articles and therefore amend this version and propose a new one.

Author Response

Response to Reviewer #1

We appreciate the reviewer’s comments. We have modified the manuscript accordingly. The main changes are highlighted in the revised version. In the following, our responses are marked point by point by point in blue.

Comments to authors:

The article is a collection of interesting results.

The authors deliver results in large quantities, but what is lacking is to highlight what these results bring back that is innovative compared to what exists in the bibliography. Authors are encouraged to compare their results with other articles and therefore amend this version and propose a new one.

R: Thank you for the suggestion. We agree with this suggestion and the novelty and significance of the present have been highlighted in the last paragraph of the introduction section as follows:

The novelty of this study is the use of full iron tailings concrete (FITC), which was produced using iron tailings powder as an admixture, iron tailings gravel as a coarse aggregate and iron tailings sand as a fine aggregate. In addition, to the best of our knowledge, this is the first report in the literature to investigate the structural behaviour of FITRC columns under large eccentric loads. To comprehensively evaluate the characteristics and limitations of FITRC columns in order to provide a theoretical basis for the large-scale use of iron tailings. We believe that this study can be of practical significance to structural engineers in China.

In the results and discussion, we added one latest relevant reference on the mechanical behaviour of iron tailings reinforced concrete specimens, and compared the results of this study with those of two relevant references.

In section 3.1( Failure modes and crack propagation) of the manuscript, the reads as follows:

The failure modes of FITRC and CRC columns under large eccentric loads are essentially similar, and the failures of all specimens were manifested by longitudinal bar yielding in the tension zone, followed by concrete crushing in the compression zone (Figure 8). The eccentricity of the FITRC45 and CRC45 columns were 0.60 and those of the FITRC35 and CRC35 columns were 0.70. In a previous study [55], the eccentric compressive behaviour of iron tailings sand RC columns was investigated and similar results were obtained.

The literature [55] reports the use of iron tailings sand only as fine aggregate. In this study, full iron tailings concrete (FITC) was developed using iron tailings. The concrete raw materials consisted of finely ground iron tailings powder as an admixture, and iron tailings gravel and sand as coarse and fine aggregates, respectively.

In section 3.3( Deformation and Ductility) of the manuscript, the reads as follows:

Table 8 shows that the mean ductility factor of the three FITRC45 and FITRC35 columns is 1.78 and 2.82, respectively, and the ductility factors relative to the CRC45 and CRC35 columns are reduced by 36.6% and 19.5%, respectively. Therefore, an increase in section size and eccentricity reduces the difference in ductility between FITRC35 and CRC35 columns. A previous study [52] examined the axial compressive properties of full iron tailings RC columns and found that the ductility coefficients of FITRC columns were also lower than those of CRC columns.

The literature [52] only emphasises the axial compressive properties of the specimens. In this study, both the compressive and flexural properties of the specimens are specifically investigated.

Reference

52.Ma, X.; Sun, J.; Zhang, F.; Yuan, J.; Meng Z. Experimental studies and analyses on axial compressive properties of full iron tailings concrete columns. Case Stud Constr Mat 2023, 18, e1881. https://doi.org/https://doi.org/10.1016/j.cscm.2023.e01881.

55.Yan, P. Experimental study on mechanical behaviour of high performance iron tailing concrete columns. MEng dissertation. Hebei University of Architecture, Zhangjiakou, China, 2017. https://doi.org/10.27870/d.cnki.ghbjz.2017.000013.

Reviewer 2 Report

The manuscript concerns the usage the full iron tailings concrete as a structural concrete in RC columns. The topic of this research is important due to the fact that the iron tailings constitute huge amount of waste during separating process of iron-rich particles. Using the iron tailings in concrete (as coarse and fine aggregate and cement replacement powder) can reduce the problem of storage of this wase material.

 The quality of the manuscript is very good. I have some minor suggestions for the Authors to consider in the revised version of the manuscript

 1. The title of the manuscript is misleading. The Authors carried their research in the short-term loading conditions (both for the concrete specimens and for the columns). Therefore, I suggest introducing in the title, and then underline it in abstract that all the findings in manuscript are pertinent to short-term loadings. The title can be changed in the following way: “Evaluating the mechanical behaviour of reinforced concrete columns comprising full iron tailings under large eccentric short-term loading”.

2. I suggest introducing in Conclusions that the FITC concrete columns was tested under short-term loading and additional research is needed in order to describe its long-term behaviour (creep, shrinkage).

3.  Page 17, equation 4. This equation is valid only if the first and second order deformation of a column can be expressed as the sinusoidal shape. This remark should be introduced in the revised version of the manuscript.

Author Response

Response to Reviewer #2

We appreciate the reviewer’s comments. We have modified the manuscript accordingly. The main changes are highlighted in the revised version. In the following, our responses are marked point by point by point in blue.

Comments to authors:

The manuscript concerns the usage the full iron tailings concrete as a structural concrete in RC columns. The topic of this research is important due to the fact that the iron tailings constitute huge amount of waste during separating process of iron-rich particles. Using the iron tailings in concrete (as coarse and fine aggregate and cement replacement powder) can reduce the problem of storage of this wase material.

The quality of the manuscript is very good. I have some minor suggestions for the Authors to consider in the revised version of the manuscript:

  1. The title of the manuscript is misleading. The Authors carried their research in the short-term loading conditions (both for the concrete specimens and for the columns). Therefore, I suggest introducing in the title, and then underline it in abstract that all the findings in manuscript are pertinent to short-term loadings. The title can be changed in the following way: “Evaluating the mechanical behaviour of reinforced concrete columns comprising full iron tailings under large eccentric short-termloading”

R1: Thank you for the suggestion. We agree with this suggestion and the title has been revised. New title “Mechanical behaviour evaluation of full iron tailings concrete columns under large eccentric short-term loading”.

The abstract has been revised. In the abstract, the following statement was added:

Therefore, eight rectangular reinforced concrete (RC) columns with conventional reinforced concrete (CRC) as a control were tested to investigate the effects of section dimensions, initial eccentricities, and concrete strengths, on the structural behaviour of FITRC columns under large eccentric short-term loading. The experimental and analytical results indicated that the sectional strain of the FITRC columns satisfied the plane-section assumption under short-term loading, and the lateral deflection curve agreed well with the half-sinusoidal curve.

  1. I suggest introducing in Conclusions that the FITC columns was tested under short-term loading and additional research is needed in order to describe its long-term behaviour (creep, shrinkage).

R2: The conclusions has been revised. In the conclusions, the following statement was added:

(conclusions 1) Under large eccentric short-term loads, the failure modes of the FITRC and CRC columns were found to be similar, and the failures were manifested by the yielding of the tensile and compressive rebars and concrete crushing in the compression zone. The long-term behaviour of FITRC (creep and shrinkage) requires further investigation.

  1. Page 17, equation 4. This equation is valid only if the first and second order deformation of a column can be expressed as the sinusoidal shape. This remark should be introduced in the revised version of the manuscript.

R3: This sentence was added in the revised version of the manuscript:

This equation is valid only if the first- and second-order deformations of a column can be expressed as sinusoidal shapes.

Reviewer 3 Report

Dear , I think that this presented study is good and deserves publication. Generally, I enclose some comments that need to be responded to before publication.  

Abstract,

1- It is preferable to highlight the purpose of the study.

2- Include the most important results that achieved the goal of the research.  

Keywords.

1- Some words are long, it is preferable to put words that attract the attention of researchers (be short)  

Introduction.

1- Add a section explaining whether the iron tailings were applied by researchers in the construction sectors.

2- What is the mechanism by which concrete is improved by using iron tailings? 

Methodology.

1- It is preferable to clarify more about the design principles of the concrete mixture and the RC columns.    

Results and discussion.

1- It is required to cite relevant references that support the results achieved in this study.  

Conclusion.

Abstracts 4, 5, and 7 are a bit long and need to be rephrased to make it easier for the reader.  

Greetings

Author Response

Response to Reviewer #3

We appreciate the reviewer’s comments. We have modified the manuscript accordingly. The main changes are highlighted in the revised version. In the following, our responses are marked point by point by point in blue.

Comments to authors:

Dear , I think that this presented study is good and deserves publication. Generally, I enclose some comments that need to be responded to before publication.

  1. Abstract. It is preferable to highlight the purpose of the study.

R1: Thank you for the suggestion. We agree with this suggestion and the abstract has been revised. In the abstract, the following statement was added:

Iron tailings account for 86.8% of the total mass of solid raw materials in the FITC. To enable large-scale use of FITC, a comprehensive investigation of the structural behaviour of full-iron tailing-reinforced concrete (FITRC) specimens is warranted.

  1. Abstract. Include the most important results that achieved the goal of the research.

R2: The abstract has been revised. In the abstract, the following statement was added:

In addition, the FITRC columns exhibited a slightly lower cracking load and ultimate load capacity than the CRC columns, and the crack widths were larger than those of the CRC columns. The reduction in the load capacity observed in the FITRC is within the permissible range stated in the design code, thereby satisfying the code requirements. The deformation coefficients of the FITRC and CRC columns were identical, and the cracking and ultimate loads calculated according to the current code and theories were in good agreement with the measured results.

  1. Keywords. Some words are long, it is preferable to put words that attract the attention of researchers (be short)

R3: Thank you for the suggestion. We agree with this suggestion and the keywords has been revised. Keywords are the following:

Iron tailings; RC column; Large eccentric; Short-term load; Bearing capacity; Ductility

  1. Introduction. Add a section explaining whether the iron tailings were applied by researchers in the construction sectors.

R4: The third section of the introduction is devoted to highlighting the current state of research efforts by construction sectors researchers in relation to iron tailings. The third section of the introduction reads as follows:

Iron tailings are inert at room temperature [26,27]. Previous studies have used iron tailings as coarse [28,29] and fine aggregates [30-35], as well as silica materials for the production of aerated concrete at high temperatures [36-38]. Some studies have also used iron tailings as an admixture [39-44] and examined the activation of iron tailings powder [45,46]. The use of iron tailings in concrete has the potential to reduce the storage of iron tailings, reduce land use and reduce environmental pollution. At the same time, the replacement of fly ash with iron tailings powder as an admixture could reduce the cost of concrete and promote the sustainable development of concrete. In addition, iron tailings can improve the carbonation resistance, frost resistance and sulphate erosion resistance of concrete [47,48].

To enable the extensive application of iron tailings in reinforced concrete, it is imperative to validate the structural behaviour of reinforced concrete samples containing iron tailings compared with plain concrete. A study was conducted to compare the flexural behaviour of iron tailings sand concrete beams and conventional concrete (CC) beams [49]. The results showed that the two types of concrete beams exhibited similar flexural behaviours and that the calculation of the crack width of iron tailings sand concrete beams should be corrected when the replacement rate of iron tailings sand is greater than 40%. Another study [50] investigated the axial compressive behaviour of short iron tailings sand concrete columns. The results showed that the iron tailings sand concrete short columns were comparable to CC short columns in terms of axial compressive strength, deformation, and ductility. The seismic behaviour of iron tailings sand concrete columns was investigated [51]. The results showed that the failure patterns of iron tailings sand concrete columns and CC columns were almost identical, while the flexural capacity of the iron tailings sand concrete columns was slightly different from that of the CC columns, and iron tailings sand could completely replace conventional sand. Compared with CC columns, the axial compressive properties of full iron tailing concrete columns satisfied the requirements of the current code [52]. Further, the calculation model that considered the hoop restraint effect could more accurately predict the axial compressive bearing capacity of full iron tailings concrete columns.

  1. Introduction. What is the mechanism by which concrete is improved by using iron tailings?

R5: The use of iron tailings in concrete has the potential to reduce the storage of iron tailings, reduce land use and reduce environmental pollution. At the same time, the replacement of fly ash with iron tailings powder as an admixture could reduce the cost of concrete and promote the sustainable development of concrete. In addition, iron tailings can improve the carbonation resistance, frost resistance and sulphate erosion resistance of concrete [47,48].

Reference

  1. Zhang R. Experimental studies of influence for iron tailings on carbonation and frost resistance of concrete. MEng dissertation. Hebei Agricultural University, Baoding, China, 2022.
  2. Yang M.; Sun J.; Xu Y.; Wang J. Analysis of influence from iron tailings powder on sulfate corrosion resistance of concrete and its mechanism. Water Resources and Hydropower Engineering. 2022, 53(11), 177-85. https://doi.org10.13928/j.cnki.wrahe.2022.11.017.

  1. Methodology. It is preferable to clarify more about the design principles of the concrete mixture and the RC columns.

R6: Thank you for the suggestion. In Section 2.2( Experiment design) of the manuscript, the design principles of the concrete mixture and the RC columns have been clarified as follows:

To reduce the effect of an additional bending moment on an eccentrically pressurised column specimen, the span-depth ratio (L/h) should not exceed 5. The 1200 mm high specimen has 150 mm × 250 mm sectional dimensions and symmetrical reinforcement of two 14 mm rebars in tension and two 14 mm rebars in compression, made of C45 strength concrete. The 1500 mm-high specimen has a 200 mm × 300 mm section, and asymmetric reinforcement with three 16 mm rebars in tension and two 16 mm rebars in compression made from C35 concrete. To ensure that the specimens could withstand a large eccentric load, cow legs were placed at the top and bottom of the specimen.

  1. Results and discussion.It is required to cite relevant references that support the results achieved in this study.

R7: In the results and discussion, The two latest relevant reference about the mechanical behaviour of iron tailings in reinforced concrete have been added and discussed.

In Section 3.1( Failure modes and crack propagation) of the manuscript, the reads as follows:

The failure modes of FITRC and CRC columns under large eccentric loads are essentially similar, and the failures of all specimens were manifested by longitudinal bar yielding in the tension zone, followed by concrete crushing in the compression zone (Figure 8). The eccentricity of the FITRC45 and CRC45 columns were 0.60 and those of the FITRC35 and CRC35 columns were 0.70. In a previous study [55], the eccentric compressive behaviour of iron tailings sand RC columns was investigated and similar results were obtained.

In Section 3.3( Deformation and Ductility) of the manuscript, the reads as follows:

Table 8 shows that the mean ductility factor of the three FITRC45 and FITRC35 columns is 1.78 and 2.82, respectively, and the ductility factors relative to the CRC45 and CRC35 columns are reduced by 36.6% and 19.5%, respectively. Therefore, an increase in section size and eccentricity reduces the difference in ductility between FITRC35 and CRC35 columns. A previous study [52] examined the axial compressive properties of full iron tailings RC columns and found that the ductility coefficients of FITRC columns were also lower than those of CRC columns.

Reference

  1. Ma, X.; Sun, J.; Zhang, F.; Yuan, J.; Meng Z. Experimental studies and analyses on axial compressive properties of full iron tailings concrete columns. Case Stud Constr Mat 2023, 18, e1881. https://doi.org/https://doi.org/10.1016/j.cscm.2023.e01881.
  2. Yan, P. Experimental study on mechanical behaviour of high performance iron tailing concrete columns. MEng dissertation. Hebei University of Architecture, Zhangjiakou, China, 2017. https://doi.org/10.27870/d.cnki.ghbjz.2017.000013.

  1. Conclusion.Abstracts 4, 5, and 7 are a bit long and need to be rephrased to make it easier for the reader.

R8: Thank you for the suggestion. We agree with this suggestion and the conclusion (4, and 5) has been revised. The 7th conclusion has been deleted. Conclusion are the following:

  1. Compared with the CRC45 and CRC35 columns, the ductility factors of the FITRC45 and FITRC35 columns were 36.6% and 19.5% lower, respectively. The underlying cause of this phenomenon is the comparatively low modulus of elasticity of FITC, which results in a more pronounced lateral deformation of FITRC columns when subjected to eccentric loading than CRC columns under equivalent conditions.
  2. Based on the current Chinese standards, the theoretical calculations for the cracking load and ultimate load capacity of FITRC columns are relatively accurate. The calculation results indicate that FITRC columns have a certain safety reserve, and that FITC has the potential for practical application in the construction sector.

Round 2

Reviewer 1 Report

The paper is ok